# Effects of Thryptophan Hydroxylase Blockade by P-Chlorophenylalanine on Contextual Memory Reconsolidation after Training of Different Intensity

**DOI:** 10.3390/ijms21062087

**Published:** 2020-03-18

**Authors:** Irina B. Deryabina, Viatcheslav V. Andrianov, Lyudmila N. Muranova, Tatiana K. Bogodvid, Khalil L. Gainutdinov

**Affiliations:** 1Laboratory of Neuroreabilitation of Motor Disorders, Institute of Fundamental Medicine and Biology, Kazan Federal University, 420000 Kazan, Russia; ira-kan@yandex.ru (I.B.D.); slava_snail@yahoo.com (V.V.A.); m.luda@mail.ru (L.N.M.); tat-gain@mail.ru (T.K.B.); 2Laboratory of Spin Physics and Spin Chemistry, Zavoisky Physical-Technical Institute of the Russian Academy of Sciences, 420000 Kazan, Russia; 3Department of Biomedical Sciences, Volga Region State Academy of Physical Culture, Sport and Tourism, 420000 Kazan, Russia

**Keywords:** 4-Chloro-DL-phenylalanine, serotonin, learning, reminder, contextual memory, reconsolidation, anisomycin, snail

## Abstract

The processes of memory formation and its storage are extremely dynamic. Therefore, the determination of the nature and temporal evolution of the changes that underlie the molecular mechanisms of retrieval and cause reconsolidation of memory is the key to understanding memory formation. Retrieval induces the plasticity, which may result in reconsolidation of the original memory and needs critical molecular events to stabilize the memory or its extinction. 4-Chloro-DL-phenylalanine (P-chlorophenylalanine-PCPA) depresses the most limiting enzyme of serotonin synthesis the tryptophan hydroxylase. It is known that PCPA reduces the serotonin content in the brain up to 10 times in rats (see Methods). We hypothesized that the PCPA could behave the similar way in snails and could reduce the content of serotonin in snails. Therefore, we investigated the effect of PCPA injection on contextual memory reconsolidation using a protein synthesis blocker in snails after training according to two protocols of different intensities. The results obtained in training according to the first protocol using five electrical stimuli per day for 5 days showed that reminding the training environment against the background of injection of PCPA led to a significant decrease in contextual memory. At the same time, the results obtained in training according to the second protocol using three electrical stimuli per day for 5 days showed that reminding the training environment against the injection of PCPA did not result in a significant change in contextual memory. The obtain results allowed us to conclude that the mechanisms of processes developed during the reconsolidation of contextual memory after a reminding depend both on the intensity of learning and on the state of the serotonergic system.

## 1. Introduction

Serotonin (5-HT) is the main neurotransmitter that plays primary roles in learning based on defensive reflexes in mollusks [1,2,3,4,5,6,7]. Serotonin is one of the most widely distributed and well-researched neurotransmitters of the nervous system. The chemical structure of serotonin is surprisingly simple, but there is inconsistency between this simplicity of the structure of serotonin and the amazing complexity and variety of its physiological effects. It has been shown that 5-HT can perform integrative functions when released into the extracellular environment [8,9,10,11,12]. It was shown that electrophysiological correlates of plasticity can be reproduced by applying 5-HT applications to the solution surrounding the central nervous system of mollusks [13,14,15,16,17,18,19,20,21]. On the other hand, the neurotoxic analogues of serotonin 5,6- and 5,7-dihydroxytryptamine (5,6- and 5,7-DHT) leading to the depletion of serotonin is widely used to study the role of the serotoninergic system in the formation of behavior and learning [1,7,22,23,24,25,26,27,28].

4-Chloro-DL-phenylalanine (PCPA) is another substance that causes depletion of serotonin in the brain by a different mechanism [29]. The basis for the biosynthesis of serotonin is the amino acid tryptophan, which is abundant in nature. PCPA inhibits the tryptophan hydroxylase enzyme, which is the first and most limiting 5-HT biosynthesis enzyme [30]. Thus, PCPA, as neurotoxins 5,6- and 5,7-DHT, causes depleted serotonin in the brain and inhibits serotonergic transmission [29,31]. Previously we carried out the study of the comparative effects of PCPA and neurotoxins 5,7-DHT (5,6-DHT) on the formation of conditioned reflex in the terrestrial snail and found that they equally disrupted learning [3].

It was shown that the administration of neurotoxins 5,7-DHT and 5,6-DHT led to a disruption in the formation of a conditioned reflex of food aversion [1] and conditioned defensive reflex of pneumostome closure in snails [3,7]. It was found that neurotoxin 5,7-DHT blocked heterosynaptic facilitation in Aplysia [2], impaired the formation of the long-term sensitization in snail [3,32,33]. In behavioral experiments it was shown that the destruction of the functioning of the serotonergic system by neurotoxin 5,7-DHT did not change the original memory, but led to impairment of the memory after re-reactivation [28,34,35]. These results demonstrated the relevance of studying the dependence of long-term memory properties on the state of the serotonergic system.

The transition of memory from short-term to long-term form is a complex process, which is called memory consolidation. The memory that is primarily labile becomes stable over time. The process of consolidation needs gene expression and new protein synthesis [36,37,38,39,40]. Consolidated long-term memory can be reorganized as a result of a reminding process in which trained animals receive the presentation of one of the components of the contextual situation of the training process [41,42,43]. The use of a protein synthesis blocker in a short time period after reminding of a previous experience causes the disappearance of consolidated memory of this experience [44,45,46,47]. The reactivation of the memory does not occur without the presenting of a reminder [48,49,50,51]. The process of repeated consolidation of a memory with reminding was called reconsolidation, which also needs protein synthesis [52,53,54,55]. Reconsolidation of a contextual memory has been shown in invertebrate animals [49,56,57,58,59,60,61]. The investigation of the memory formation process and memory reconsolidation in mollusks is also attractive due to the possibility of investigate the correlations between memory mechanisms and changes occurs at the cellular and receptor level [40,62,63,64,65,66,67,68,69,70,71,72,73,74,75].

In present it has been established that the processes of formation and storage of memory are extremely dynamic [76]. Therefore, elucidating the nature and temporal changes that accompany encoding, storage and retrieval is a key to understanding memory formation [48]. Retrieval or reminding induces the plasticity, which may result in reconsolidation of the original memory and needs critical molecular events to stabilize the memory [50]. So, there is the question about the existence of temporary or other “windows” within which memory reconsolidation is possible. In other words retrieval can change memory only under specific conditions. Memory can be reactivated and become labile for a period of minutes to hours and then is reconsolidated to maintain long-term memory [77].

Recently, we found that the forgetting of a previously developed conditioned reflex to the situation did not occur when reminding simultaneously combined with the inhibition of protein synthesis and administration of PCPA, an inhibitor of tryptophan hydroxylase, which is a limiting enzyme of the intermediate stage of serotonin synthesis [34,78]. In the present work we investigated the process of reconsolidation of contextual memory using two different protocols of different intensity for developing a situational reflex, which, in our opinion, and also according to the literature [76], can allow forming a “memory of various strengths”. In these experiments, we used the tryptophan hydroxylase blocker PCPA and the protein synthesis blocker anisomycin (AN).

## 2. Results

The possibility of reconsolidation of a contextual memory of different strengths (acquired as a result of training according to protocols with trainings of different intensities) was investigated in conditions of the inhibition of protein synthesis and inhibition of enzyme of the intermediate stage of serotonin synthesis by injection of PCPA. The conditioned reflex in animals was developed according to the contextual paradigm “on the ball”. The process of memory reconsolidation was initiated by the procedure of “reminding” the situation on the 6th day after the completion of the training procedure (Figure 1), which consisted of placing the animals for 20 min in a training context of “on a ball”. The reminder procedure was performed 4 days after injection of PCPA. This time period was chosen based on the data that the concentration of 5-HT in the brain of rats after administration of PCPA reached a minimum by the third or fourth day and remained lowered for at least a week [79,80].

### 2.1. Reconsolidation after Training According to Protocol 1

In the first experimental series, the animals of all groups (*n* = 28) were trained according to the protocol 1 with the presentation of five electrical stimulations per day for 5 days. A significant increase of the defensive reaction (*p* < 0.001) of the defensive ommatophores retraction in the training context of “on a ball” in response to tactile stimulation confirmed the development of a situational reflex (Figure 2, tests T2 relative to T1). Testing the same snails on a flat surface also showed an increase in defensive reactions up to two times. However, the defensive reactions of animals in the context of “on a ball” increased up to 20 times after training. Comparison of the levels of defensive responses of animals in the context of “on a ball” and on a flat surface gave evidence of contextual memory, as in the case of previous studies [47,60,78]. After that we investigated the possibility of reconsolidation of contextual memory on the situation in all 4 groups of animals of this series (Figure 1).

The results of testing the defensive reaction in animals of the control group 4: “protocol 1 + Rem + SS” (*n* = 5) showed that contextual memory remained for at least 10 days (Figure 2). These animals received a reminding procedure and then injection of saline solution (SS) after training (on the 13th day of the experiment). The increased level of defensive reactions of these animals in the context of “on a ball” at the end of the experiment (67%, T7) was significantly (*p* < 0.001) different from the level of defensive reactions on a flat surface (2%, T7) and from the initial values obtained before training (2%, T1). Thus, reminding the learning context and the subsequent injection of SS did not lead to impaired memory in animals of this group.

Testing the animals of group 3: “protocol 1 + Rem + AN” (*n* = 8) trained according to the protocol 1 and received reminding the situation and subsequent injection of a protein synthesis blocker, showed a significant decrease in the level of defensive reaction in context of “on the ball” after the 14th day of the experiment. The level of the defensive reaction in a context of “on a ball” significantly (*p* < 0.001) decreased from 57% (T2) to 17% (T3), and to 13% (T7; *p* < 0.001) on the last day of testing. Thus, the procedure of reminding the situation and the subsequent injection of a protein synthesis blocker led to almost complete amnesia in animals of this group. These results indicated the presence of reconsolidation of contextual memory, which required the protein synthesis [47,60,78] and correlated with other data [44,45,48,49,52,53,54,55,61].

Testing the animals of group 2: “protocol 1 + PCPA + Rem” (*n* = 5) showed a decrease in contextual memory two times. These animals received an injection of PCPA after training according to protocol 1 and then they received the reminding procedure on the 13th day of the experiment without a subsequent injection of a protein synthesis blocker. The level of the defensive reaction in a context of “on a ball” significantly (*p* < 0.01) decreased from 78% (T2) to 40% (T3), and to 38% (T7) (*p* < 0.01) on the last day of testing.

Testing the animals of group 1 “protocol 1 + PCPA + Rem + AN” (*n* = 10) trained according protocol 1 and received a PCPA injection before reminding also showed a decrease in memory. These animals received the injection of a protein synthesis blocker after reminding procedure. The level of defensive reaction in context of “on a ball” significantly (*p* < 0.05) decreased from 71% (T2) to 45% (T3), and to 39% (T7; *p* < 0.01) on the last day of testing.

Thus, the results of this experimental series showed that a reminding the training context after training and injection of PCPA without the use of a protein synthesis blocker led to a decrease in the level of the defensive reaction in snails by about 50% of the initial value. This result indicated a weakening of contextual memory. At the same time, a reminder of the training context combined with a protein synthesis blockade, after injection of PCPA, did not result to an additional weakening of the contextual memory.

The results of this experimental series also showed that reminding the training context, followed by injection of a protein synthesis blocker, led to an almost complete violation of the contextual memory in snails, i.e., the process of memory reconsolidation, triggered by a reminding the context, was disrupted by the blockade of protein synthesis.

### 2.2. Reconsolidation after Training According to Protocol 2

All groups of the animals of the second experimental series were trained according to protocol 2 with the presentation of three electrical stimulations per day for 5 days. Testing the level of the defensive response confirmed the successful development of a situational reflex as in case with protocol 1 (Figure 3, T2 tests relative to T1).

Testing the animals of group 8: “protocol 2 + Rem + SS” (*n* = 5) trained according to protocol 2 showed that the memory in this case was preserved for at least 10 days (Figure 3). Animals of this group received the reminding procedure on the 13th day and injection of the saline solution as a control after training. The increased level of defensive reactions of these animals in the context of “on a ball” at the end of the experiment (47%, T7) significantly (*p* < 0.0001) differed from the level of defensive reactions on a flat surface (7%, T7) and from the initial values obtained before training (6%, T1). Thus, reminding the learning context and the subsequent injection of saline solution did not lead to an impaired memory in animals of this group.

Testing the animals of group 7 “protocol 2 + Rem + AN” (*n* = 5) showed that on the 14th day of the experiment there was no impairment of the memory. On the 13th day of the experiment after the training according protocol 2, animals of this group received the reminding procedure and injection of the blocker of protein synthesis AN. The level of the defensive reaction in the context of “on a ball” decreased from 51% (T2) to 36% (T3). Thus, in the case of training according to the protocol 2, the reminding the learning context and the subsequent injection of a protein synthesis blocker did not lead to memory impairment in the animals of this experimental group.

Testing the animals of group 6: “protocol 2 + PCPA + Rem” (*n* = 6) and group 5: “protocol 2 + PCPA + Rem + AN” (*n* = 5) did not show a significant decrease in the contextual memory (Figure 3). These animals received reminding of the context of training with a subsequent injection of PCPA on the 13th day of the experiment. Animals of group 5 received an injection of a protein synthesis blocker AN after reminding. The level of a defensive reaction in the context of “on a ball” reliably decreased in both groups from an average level from 64% (T2) to 42% the day after the reminding (T3). On the last day of testing (T7), the levels of defensive reaction in the context of “on a ball” were 46% (group 5) and 52% (group 6), and these changes were not significant. Thus, there was no significant decrease in the contextual memory at the end of the experiment in these two groups. Testing (T7) the levels of the defensive response on the ball in these two groups in the end of the experiment showed that there was no significant decrease in the contextual memory.

Thus, the results of the second experimental series performed using the training protocol with a lesser number of electrical stimuli (protocol 2) showed that reminding the training context against the background of reduction of serotonin by PCPA did not lead to a reliable change in the contextual memory in snails, as with a combination of reminding with both a protein synthesis blockade and without a protein synthesis blockade. The fact that the blockade of protein synthesis after reminding did not lead to a serious impairment of the memory acquired during training directly indicated a weaker dependence of the process of reconsolidation of memory on the synthesis of a new protein on training protocol 2. That is, the state of memory liability initiated by the reminding was not completed enough in this case.

### 2.3. Testing Procedures in the Context of “on the Ball” after Inhibition of Serotonin Synthesis

The third experimental series was carried out in order to check whether the testing procedure of the defensive reaction of the animals (T3–T7) used in the first two experimental series triggered the initiation of memory reconsolidation, which depended on the level of 5-HT in animals. Animals of group 9 “protocol 1 + PCPA” (*n* = 4) were trained according to protocol 1 and then they received the injection of PCPA. Animals of group 10 “protocol 1 + SS” (*n* = 5) were trained according to the same protocol and then they received the injection of saline solution as a control (Figure 4A). After that, animals of these groups were tested for defensive reactions in both contexts (on a ball and on a flat surface) at the same time as the groups from the first two experimental series. Testing the level of the defensive reactions was performed in the same context of “on a ball” as the reminding and training and without a separate reminding procedure. During testing in one of the contexts, the animals were in the situation “on the ball”, which were the same as the reminding or the training context. At the same time, the animals of these groups did not have a separate reminding session. The testing procedure was standard, the same as in the first two experimental series. It was shown that the level of defensive reactions of the ommatophores retraction in response to tactile stimulation of animals after injection of PCPA (group 9) was not changed throughout all test days (Figure 4B, T3–T7 tests).

These results indicate that the procedure of testing animals in the context of “on a ball” did not lead to the initiation of memory reconsolidation, which depends on the injection of PCPA. That is, that all changes in the level of defensive reactions of animals demonstrated in the first two experimental series did not depend on the testing procedure similar with the reminding procedure of the learning environment.

## 3. Discussion

Serotonin is a neurotransmitter that acts as a biochemical messenger and regulator in the brain. It participates in a number of important physiological functions [81,82]. The participation of the serotonin system in animal cognition and behavior, in learning and memory attracts increasing attention [28,83,84,85]. Therefore, the deregulation of the serotonergic system causes the pathogenesis of many psychiatric and neurological disorders [84,86]. These include, first of all, anxiety, depression and the phenomena referred to as serotonin syndrome [87,88,89,90]. There is a variety of evidence that links depression to a decrease in the serotonergic system activity and supports the hypothesis that changes in the 5-HT neuron content may play a role in the pathophysiology of depression [91,92]. Besides, certain clinical therapies for treating depression are based on a change in the level of 5-HT in the body [90,93].

There are a number of techniques for reducing the level of 5-HT. These are the use of 5-HT transporter blockers [85,90,94,95] and application of technologies associated with the exposure to 5-HT receptors [85,90,96,97]. Tryptophan hydroxylase, an enzyme that limits the rate of 5-HT synthesis, which regulates the activity of 5-HT in the brain, plays an important role [97,98,99,100,101]. So, by whole brain biochemical assays using high-pressure liquid chromatography with electrochemical detection it was found that the administration of PCPA at a dose of 1000 mg/kg reduced whole rat brain levels of 5-HT and its metabolite 5-hydroxyindoleacetic acid to 9.4% and 8.2% of control levels, respectively [99]. Other authors by immunohistochemical methods also showed a decrease in serotonin up to 10% from the initial level [30], PCPA at a dose of 300 mg/kg causes a decrease in serotonin in the animal up to 10 times [79]. Experimental data show that PCPA, which inhibited an enzyme of tryptophan hydroxylase, reduces not only the level of 5-HT, but also the level of catecholamines [100,102,103]. However, if the level of serotonin decreases to 10%–15% of the initial level, then the level of dopamine and norepinephrine decreases only to 85% [102,104].

Interesting comparative data on changes in the content of serotonin and catecholamines in different parts of the brain of rats using high performance liquid chromatography (HPLC) were obtained by Reader et al. (1984; 1986). It has been shown that administration of PCPA at a dose of 2 × 400 mg/kg causes a decrease in serotonin levels by 98.7%, dopamine by 96%, norepinephrine by 23.5%, and adrenaline increase by a 331% in the rat cortex [105]. Another study examined the comparative effects of neurotoxins 5,7-DHT and 6-OHDA with effects of PCPA. It was found that under the action of PCPA, serotonin levels decrease to 10–20% of the initial level, and catecholamines decrease to 50%–80% [29]. The effects of 5,7 DHT and PCPA on serotonin and catecholamine levels were similar [29,106]. However, in the nervous system, the functions of serotonin and catecholamines intersected only partially. So, it was found that PCPA completely blocks the decrease in immobilization caused by a selective serotonin reuptake inhibitor in a suspension test in mice [31]. In contrast, PCPA did not affect the effects of a norepinephrine reuptake inhibitor.

Our experiments were conducted on a mollusk—a terrestrial snail. A characteristic feature of mollusks is that their defensive reflexes are mediated by serotonin [1,2,3,4,5,8,10,85,107]. Thus, it was found that anxious-like behavior in crayfish is controlled by serotonin [108], and in the snail, the formation of long-term sensitization is blocked by the neurotoxin 5,6-dihydroxytryptamine [27,109,110]. This also applies to the conditioned reflex to the situation (context), the base for which is the serotonergic neurons of the pedal ganglion [11,65,107]. Our studies involve the use of PCPA, which inhibited an enzyme of tryptophan hydroxylase, the first and most limiting enzyme in serotonin biosynthesis. For the reasons mentioned above, we focused on the role of serotonin in the reconsolidation of the conditioned reflex to the context.

In present studies we carried out the investigation of the effect of reducing serotonin levels in the process of reconsolidation of contextual memory in animals after training of different intensities, using the protein synthesis blocker AN and blocker of enzyme tryptophan hydroxylase PCPA (Figure 5). It was found that reminding the learning environment (placing the snails in a training context of “on a ball” for 20 min) after injection of PCPA in snails trained according to protocol 1 (presenting five stimuli per day for 5 days) led to a significant (approximately two times) weakening of contextual memory. The result showed a violation of memory reconsolidation with reduced serotonin levels (Figure 2). It could be assumed that serotonin in this case was necessary for the initiation of reconsolidation, or for its complete completion. A very interesting result was that this weakening of memory in animals with a reduced serotonin level did not depend on the combination of reminding with a protein synthesis blockade.

Thus, blockade of tryptophan hydroxylase by p-chlorophenylalanine after reminding the training context in animals with a simultaneous blockade of protein synthesis did not result to additional weakening of contextual memory. This fact, might reflect either the priority of the effects of reducing serotonin content by the use of PCPA in the phenomenon of contextual memory reconsolidation in the snail as the injection of AN did not affect in the situation of the disruption of the serotonin system, or the complex mutual effect of reducing the serotonin content by PCPA and blocking protein synthesis on reconsolidation, which led to its violation and reduced a trace of memory. The reason for the absence of the cumulative effect of the blockade of protein synthesis and serotonin synthesis is probably the fact that a long-term increase in the 5-HT level in hemolymph was blocked by an inhibitor of protein synthesis [111].

In the case of training according to a different protocol (presentation of three stimuli per day for 5 days), the reminding of the learning environment with the subsequent injection of a protein synthesis blocker did not result in such a significant forgetting of the contextual memory. In this case, the use of PCPA also did not lead to a significant decrease of memory in both contexts with the blockade of protein synthesis and without it. At the same time, the reminding procedure with the subsequent injection of a biosynthesis blocker AN without injection of PCPA did not lead to such a significant disruption of the contextual memory reconsolidation and to its almost complete loss.

The obtained results possibly indicated that the use of PCPA, which presumably reduced the level of serotonin in the nervous system of the snail, might partially block the reminding, which was necessary to start the reconsolidation process. Perhaps for this reason, the impairment of protein synthesis during the reminding in group 1 did not cause a complete blockade of reconsolidation of contextual memory into a situational conditioned reflex. On the other hand, for the same reason, in the case of the use of PCPA in animals of experimental groups without a blockade of protein synthesis by anisomycin, the reminding procedure after depletion of the serotonin level by PCPA could also lead to incomplete reconsolidation. However, these assumptions need to be verified. The obtain results could be evidence that a serotonergic system was included in the process of memory reconsolidation (in our system of contextual memory). Mechanisms of processes that developed during the reconsolidation of contextual memory after a reminder depended on the 5-HT system and on the state of the memory at the time of the reminding procedure associated with the intensity of training.

## 4. Materials and Methods

### 4.1. Experimental Animals

All experimental (behavioral) procedures are in compliance with the Guide for the Car and Use of Laboratory Animals published by the National Institutes of Health, Directive 2010/63/EU of the European Parliament and of the Council of 22 September 2010 and in accordance to guidelines of our University. For the experiments the terrestrial snails *Helix lucorum* (Gastropoda, Pulmonata), were used. The capture of animals in the wild were carried out by competent persons without avoidable pain and distress (Article 9 of Directive 2010/63/EU). Snails transported asleep and then most of them were also stored asleep (Article 33 of Directive 2010/63/EU). Prior to the experiments the snails were kept for no less than two weeks in a glass terrarium in a humid atmosphere at room temperature (each group in a separate terrarium) (Article 33 of Directive 2010/63/EU). All groups were housed in separate terrariums which were kept together all the time in the same room under the same conditions. After behavioral experiments snails were outputted again into the terrarium. Snails were kept in the active state for at least 2 weeks before experiments with the possibility of free movement. Snails of approximately the same weight (about 25 g) were selected for experiments. The animals were deprived of food for three days prior to the experimental sessions.

### 4.2. Drugs and Injections

AN (anisomycin (2-[p-Methoxybenzyl]-3,4-pyrrolidinediol 3-acetate, Sigma) was used for a blockade of the protein synthesis [112] and PCPA (4-Chloro-DL-phenylalanine methyl ester hydrochloride 97%, Sigma) was used for a blockade of serotonin synthesis.

4-Chloro-DL-phenylalanine inhibits the tryptophan hydroxylase enzyme, which is the first and most limiting enzyme of 5-HT biosynthesis [30,98,113]. PCPA causes a decrease in serotonin in the animal up to 10 times [29,30,31,99,102,105]. It was found, that intraperitoneal injection of PCPA in dose 100–300 mg/kg induces a dose dependent decrease of cortical content of 5-HT 3 times in 24 h after injection and 6–9 times in 2–4 days after injection [79,101,113]. Other researchers showed that intraperitoneal injection of PCPA in a dose of 300 mg/kg led to a gradual decrease of 5-HT in the brain of rats, reached maximum on the third day after injection and was maintained for at least one week [80].

Previously we found that PCPA at doses mentioned above induced the impairment of the formation of defensive conditioning in terrestrial snails as neurotoxins 5,7-DHT and 5,6-DHT [3], which occurred in the condition of the serotonin depletion in the nerve system [1,2,3,7,27,28,77,114,115].

Anisomycin dissolved in 0.2 mL of physiological saline solution (SS) was injected at a dose of 16 mg/kg of animal weight (0.4 mg/per snail). PCPA dissolved in 0.1 mL of SS was injected at a dose of 200 mg/kg animal weight (8 mg/per snail). Injections were performed by a thin needle through the insensitive part of the snail skin in the region of the sinus node [47,78].

### 4.3. Contextual Learning

The situational conditioned reflex was developed in all experimental animals according to the contextual paradigm of “on the ball” in a situation in which the shell of animals was rigidly attached to a tripod in one position, but at the same time snails had freedom of movement on the surface of the ball floating in the water. The training consisted of presenting an unconditioned stimulus (electrical stimulation) when the snail was in the specific context of “on the ball”. The training was carried out according to 2 protocols. The first protocol (protocol 1) included the presentation of 5 electrical stimulations per day for 5 days by touching two electrodes to the back and front dorsal part of the leg [47]. The second protocol (protocol 2) included 3 electrical stimulations per day for 5 days. The electrical stimulation was carried out with the following parameters: rectangular current pulses with a frequency of 50 Hz, current strength of 1–2 mA and the duration of stimulation of 1 s. The time intervals between stimuli were about 15–20min. The intensity of the stimulation current was selected to trigger a defensive reaction associated with retraction of the front part of the body and did not exceed 2 mA. The used current did not cause damage to the skin of animals, which can occur as forming of pigmented areas after stimulation with a larger current value [116,117].

The training procedure lasted 5 days, during training the snails did not receive food. The deprivation of food in invertebrate animals during the developing of conditioned reflexes is a standard technique, it is not associated with the metabolism of certain substances, but is determined by the need for an active state of the animal [28,60,61,78]. The procedure for “reminding” the training environment was to place the animals for 20 min in a training context of “on a ball”. At the same time, the animals were rigidly attached to a tripod and had freedom of movement on the surface of a ball floating in the water, as in the training context, but did not receive either tactile or electrical simulations.

### 4.4. Testing

The level of the defensive reaction of animals was tested in two contexts the day before the start of the development of the conditioned reflex, after training and in the process of further manipulations. The test was included the measurement of the amplitude of ommatophores retraction in response to mild tactile stimulation, which was a sliding movement of the brush hair with the standard speed on the skin dorsally from the front part of a snails foot. The hair touched the skin of the animal for about 1 cm and moved at an approximate speed of 1 cm/s. The maximal length of retraction of ommatophores was taken as 100%. Testing was carried out in two environments (contexts): on a flat surface (a glass cover of the aquarium) and in the training context of “on the ball” [57,78]. Each test consisted of 5 tactile stimulus presentations (intervals between tests were 7–10min). At first testing carried out on a flat surface and then animals were transferred on a ball. Tests were conducted visually and recorded on video.

Testing of the initial level of the defensive reaction was carried out the day before the development of the conditioned reflex according to the contextual paradigm (T1). The animals rested for the one (first) day after 5 days of training. On the 2nd day after training testing were repeated (T2) to confirm the development of a conditioned reflex to the situation. The situational conditioned reflex was considered developed in the case of a significant increase in the level of the defensive reaction of animals in a training context of “on the ball” in comparison to the initial testing. In the following days, the level of the defensive reaction of animals was tested at different stages of the experiment as an indicator of the preservation of long-term memory (after injection sessions and reminders of the situation T3–T7 tests).

### 4.5. Experimental Groups

Three series of experiments with 10 experimental groups of animals were carried out. Animals of the first experimental series (*n* = 28) were trained according to the protocol 1 with the presentation of 5 electrical stimulations per day for 5 days when the animals were in a training context of “on the ball”. The animals of the second experimental series (*n* = 21) were trained according to protocol 2 with the presentation of 3 electrical stimulation per day for 5 days when the animals were in the same context of “on the ball”. Thus, the first and second series of experiments differed from each other only in the protocol of training, namely, in the number of presentations of electrical stimulations to animals during the development of the situational conditioned reflex. The general procedure scheme was shown in Figure 1. The experiments began (day 1) with testing the defensive reaction of ommatophores retraction in response to tactile stimulation in animals of all groups in two contextual environments “on the ball” and on a flat surface (see above). The situational conditioned reflex was developing for 5 days according to protocol 1 or protocol 2 (from 2nd to 6th days) and then animals had a rest day (day 7). Testing the level of the defensive response of animals in both environments was repeated on the 8th day. On the 9th day, the animals of both experimental series were divided into 2 subgroups. In one subgroup, the animals received an injection of PCPA, and animals of the other did not receive any injections. From the 10th to the 12th day, it was a period of rest days, which was associated with the expectation of a maximum decrease of 5-HT in the organism of snails after the injection of PCPA [79,80,113]. Then, on the 13th day both subgroups were further divided on the following 2 subgroups. In the subgroup after the injection of PCPA the animals were further divided on groups by so: the 1st subgroup: reminding the context with injection of anisomycin or the 2nd subgroup: “reminding” the context without an injection of anisomycin. The subgroup without an injection of PCPA was divided into groups: the 3rd subgroup: “reminding” with injection of anisomycin after “reminding” and the 4th subgroup: “reminding” group with injection of the saline solution after “reminding” as a control. Thus, there were 4 groups in each of these two experimental series. Finally, over the next 5 days (from the 14th to the 18th day) completing the experimental series, the testing the level of the defensive reaction of animals in both contexts (“on the ball” and on a flat surface) was repeated. The animals of each group were trained to the situational conditioned reflex and then received a “reminding” (“Rem”) about the environmental context in which they were trained. The difference between the experimental groups of animals was in the training protocols (protocol 1 or protocol 2) and in the used pharmacological agents.

Eight groups of animals participated in first two experimental series. Groups of animals of the first experimental series were trained according to the protocol 1, with 5 electrical stimulations per day for 5 days. The animals belonged to the group 1 “protocol 1 + PCPA + Rem + AN” (*n* = 10) were injected by PCPA after training and received a reminding the training context with the subsequent injection of a biosynthesis blocker AN. The animals belonged to group 2: “protocol 1 + PCPA + Rem” (*n* = 5) were injected by PCPA after training and received reminding procedure without injection of the biosynthesis blocker AN. The animals received a reminding procedure and then injection of a protein synthesis blocker after training belonged to group 3: “protocol 1 + Rem + AN” (*n* = 8). The animals received reminding procedure and then injection of the saline solution as the control belonged to group 4: “protocol 1 + Rem + SS” (*n* = 5).

The corresponding groups of animals of the second experimental series were trained according to protocol 2. There were group 5 “protocol 2 + PCPA + Rem + AN” (*n* = 5); group 6 “protocol 2 + PCPA + Rem” (*n* = 6); group 7 “protocol 2 + Rem + AN” (*n* = 5) and group 8 “protocol 2 + Rem + SS” (*n* = 5).

The third experimental series was conducted in order to check whether the testing procedure on the ball itself, which is similar to the reminding procedure, led to the initiation of memory reconsolidation, presumably depended on the level of 5-HT in the animal body. During testing the state of the defensive system, as well as during “reminding”, the animals were placed on the ball, in the environmental context in which the training took place. However, during the testing the animal represented the active response after receiving the tactile stimulation, which was not similar to the “reminding” procedure. In addition, the testing procedure was longer and consisted of 5 presentations of the tactile stimulus with intervals between them. The general scheme of procedures for the third experimental series is shown in Figure 4A.

### 4.6. Statistics

The results are shown as mean ± SEM. The paired and unpaired Student’s *t*-test and signed rank test were used for comparison within the each experimental group. Statistical software SigmaPlot v11 was used. The statistical significance criterion was *p* < 0.05.

## Highlights

A blockade of tryptophan hydroxylase by p-chlorophenylalanine led to a weakening of contextual memory after reminding the situation of trained animals.A blockade of tryptophan hydroxylase by p-chlorophenylalanine did not lead to a significant change in contextual memory after reminding of the situation to animals learned by lesser intensity.

## Figures and Tables

**Figure 1 ijms-21-02087-f001:**
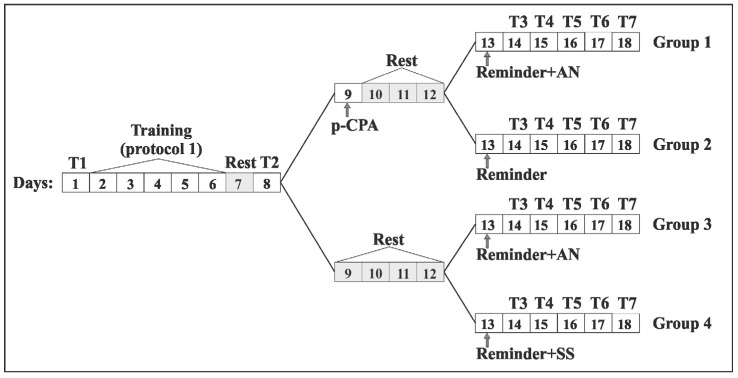
Scheme of the experimental series 1 (protocol 1) and series 2 (protocol 2; Groups 1/5, 2/6, 3/7 and 4/8). In the boxes indicated the days of the experiment. 1 day (T1)—initial testing of the level of defensive reactions, 2–6 days (Training)—development of a conditioned reflex in the context paradigm, Day 7 (Rest)—rest, Day 8 (T2)—testing of the level of defensive reaction after the procedure of training. Groups 1/5 and 2/6 “p-CPA + reminder + AN” and “p-CPA + reminder”: Day 9—injection of p-CPA, 10–12 days (Rest)—rest, 13 day Group 1/5: (Reminder + AN)—reminder session and subsequent injection of anisomycin (AN), Group 2/6: (Reminder)—reminder session, 14–18 days (T3, T4, T5, T6, T7)—testing of the level of defensive reaction after reminder session. Groups 3/7 and 4/8 “reminder + AN” and “reminder + SS”: 9–12 days (Rest)—rest, 13 day Group 3: (Reminder + AN)—reminder session and subsequent injection of anisomycin (AN), Group 4: (Reminder)—reminder session and subsequent injection of saline solution (SS), 14–18 days (T3, T4, T5, T6 and T7)—testing of the level of the defensive reaction after a reminder session.

**Figure 2 ijms-21-02087-f002:**
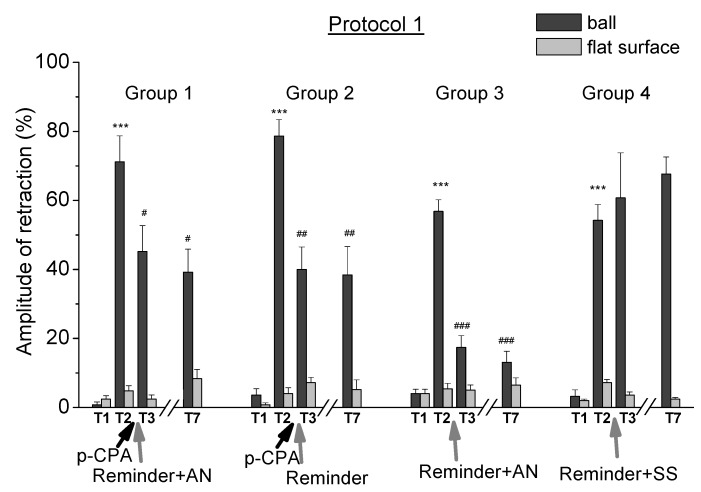
The level of the defensive response (the amplitude of the response of ommatophores withdrawal) of snails in two contexts, on the ball and flat surface for the experimental series 1 (protocol 1; Groups 1, 2, 3 and 4). T1—initial testing before the beginning of training, T2—testing 2 days (after 1 day of rest) after elaboration of conditioned reflex (learning), T3–T7—testing of animals after the injections of PCPA, AN and a reminder on the 4th–8th day after areminder (see Figure 1). Arrows indicate: PCPA—time of injection of PCPA (3 days beforethe reminder and injection of AN), Reminder—time of reminder, AN—time of injection of AN. Asterisks and hash signs indicate significant difference of the amplitude of response of ommatophores withdrawal in responses to T3–T7 vs. the amplitude of the response of ommatophores withdrawal in responses to T2 by a paired *t*-test (^#^) equal to *p* < 0.05; (^##^) equal to *p* < 0.01; (^###^) equal to *p* < 0.001 and T2 vs. amplitude of response of ommatophores withdrawal in responses to T1 by paired *t*-test (***) equal to *p* < 0.001. Vertical axis shows value of defensive reaction in response to conditioned stimulus (the amplitude of reaction of ommatophores withdrawal), in % to maximum. Horizontal axis shows the course (protocol) of the experiment: T1, T2, T3–T7, PCPA, AN and Reminder.

**Figure 3 ijms-21-02087-f003:**
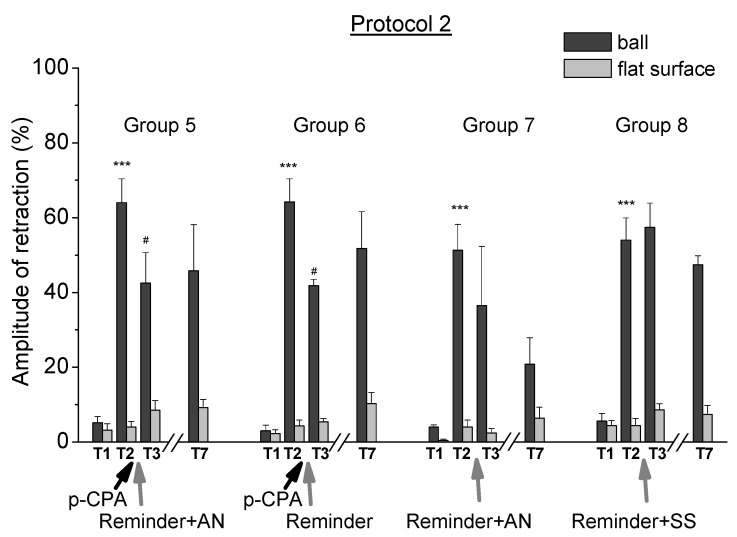
The level of the defensive response (the amplitude of response of ommatophores withdrawal) of snails in two contexts, on the ball and flat surface for the experimental series 2 (protocol 2; Groups 5, 6, 7 and 8). T1—initial testing before the beginning of training, T2—testing 2 days (after 1 day of rest) after elaboration of conditioned reflex (learning), T3–T7—testing of animals after the injections of PCPA, AN and a reminder on the 4th–8th day after a reminder (see Figure 1). Arrows indicate: PCPA—time of injection of PCPA (3 days before the reminder and injection of AN), Reminder—time of reminder, AN—time of injection of AN. Asterisks and hash signs indicate significant difference of the amplitude of response of ommatophores withdrawal in responses to T3–T7 vs. the amplitude of the response of ommatophores withdrawal in responses to T2 by a paired *t*-test (^#^) equal to *p* < 0.05 and T2 vs. amplitude of response of ommatophores withdrawal in responses to T1 by a paired *t*-test (***) equal to *p* < 0.001. The vertical axis shows the value of the defensive reaction in response to the conditioned stimulus (the amplitude of reaction of ommatophores withdrawal), in % to maximum. Horizontal axis shows the course (protocol) of the experiment: T1, T2, T3-T7, PCPA, AN and Reminder.

**Figure 4 ijms-21-02087-f004:**
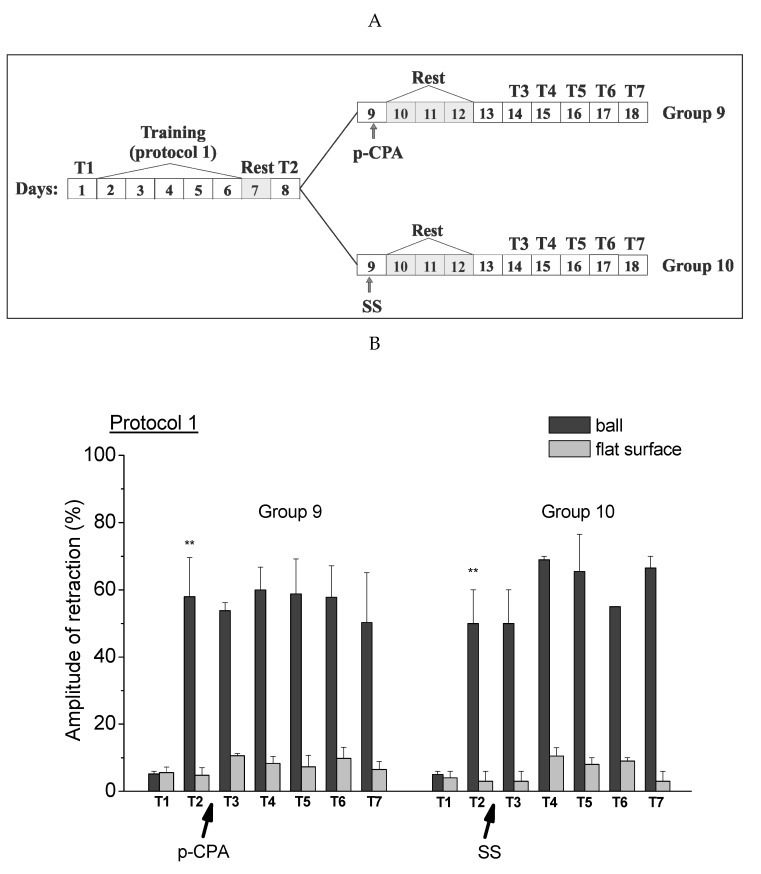
**A.** Scheme of the experimental series 3 (protocol 1; Groups 9 and 10). In the boxes indicated the days of the experiment. 1 day (T1)—initial testing of the level of defensive reactions, 2–6 days (Training)—development of the conditioned reflex in the context paradigm, Day 7 (Rest)—rest, Day 8 (T2)—testing of the level of the defensive reaction after the procedure of training. Group 9 “PCPA” and group 10 “SS”: Day 9—injection of PCPA/SS, 10–12 days (Rest)—rest, 13 day (Rest)—rest, 14–18 days (T3, T4, T5, T6 and T7)—testing of the level of the defensive reaction after the reminder session. **B.** The level of the defensive response (the amplitude of the response of ommatophores withdrawal) of snails in two contexts, on the ball and flat surface for the experimental series 1 (protocol 1; Groups 9 and 10). T1—initial testing before the beginning of training, T2—testing 2 days after elaboration of the conditioned reflex (learning), T3–T7—testing of animals after the injections of PCPA and SS on the 4th–8th day after reminder (see Figure 4A). Arrows indicate: PCPA—time of injection of PCPA, SS—time of injection of SS. Asterisks indicate significant difference of the amplitude of response of ommatophores withdrawal in responses to T2 vs. the amplitude of the response of ommatophores withdrawal in responses to T1 by a paired *t*-test (**) equal to *p* < 0.01. Vertical axis shows value of defensive reaction in response to conditioned stimulus (the amplitude of the reaction of ommatophores withdrawal), in % to maximum. Horizontal axis shows the course (protocol) of the experiment: T1, T2, T3–T7, PCPA, AN and Reminder.

**Figure 5 ijms-21-02087-f005:**
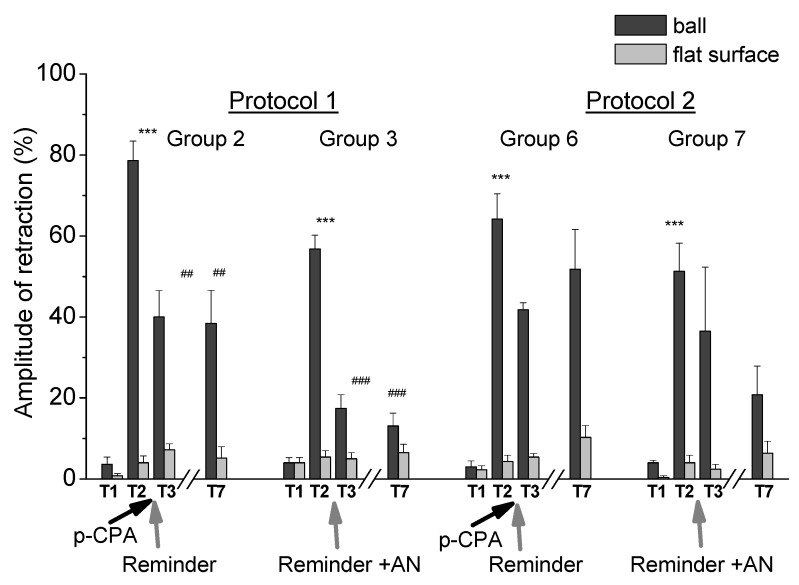
The level of the defensive response (the amplitude of the response of ommatophores withdrawal) of snails in two contexts, on the ball and flat surface for the experimental series 1 (protocol 1; Groups 2 and 3) and experimental series 2 (protocol 2; Groups 6 and 7). T1—initial testing before the beginning of training, T2—testing 2 days after elaboration of the conditioned reflex (learning), T3–T7—testing of animals after the injections of PCPA, AN and a reminder on the 4th–8th day after a reminder. Arrows indicate: PCPA—time of injection of PCPA (3 days before a reminder and injection of AN), Reminder—time of reminder, AN—time of injection of AN. Asterisks and hash signs indicate a significant difference of the amplitude of the response of ommatophores withdrawal in responses to T3–T7 vs. the amplitude of the response of ommatophores withdrawal in responses to T2 by a paired *t*-test (^##^) equal to *p* < 0.01; (^###^) equal to *p* < 0.001 and T2 vs. the amplitude of the response of ommatophores withdrawal in responses to T1 by a paired *t*-test (***) equal to *p* < 0.001. The vertical axis shows the value of the defensive reaction in response to a conditioned stimulus (the amplitude of the reaction of ommatophores withdrawal), in % to maximum. Horizontal axis shows the course (protocol) of the experiment: T1, T2, T3–T7, PCPA, AN and Reminder.

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
