# Peer review of "Effects of Thryptophan Hydroxylase Blockade by P-Chlorophenylalanine on Contextual Memory Reconsolidation after Training of Different Intensity"

_ijms, 2020, doi:10.3390/ijms21062087_

Round 1

Reviewer 1 Report

Authors replied to previous comments and I have no further questions. For future studies, I think they will greatly improve the impact of their findings by measuring 5-HT levels in snails.

Conclusions are supported by presented results. However, a double check for english language is recommended (typos and or grammar mistakes are sometimes present. e.g. did not resulted at line 323).

Author Response

Effects of tryptophan hydroxylase blockade by p-chlorophenylalanine on contextual memory reconsolidation after training of different intensity

Authors

Irina Deryabina, Viatcheslav Andrianov, Lyudmila Muranova, Tatiana Bogodvid, Khalil Gainutdinov*

Dear Reviewer 1,

Thank You for the thorough review. We consider Your comments.

  1. We have reduced the beginning of the discussion since it is not directly related to the work performed by us.
    2. In the final part of the discussion, we added an attempt to explain what processes change in response to memory reactivation during serotonin deficiency. (It could be assumed that serotonin in this case was necessary for the initiation of reconsolidation, or for its complete completion. The absence of the cumulative effect of blocking protein synthesis and serotonin synthesis might indicate different mechanisms (or ways) of consolidation and reconsolidation.)
  2. We replaced sentence «molecular processes” on(Mechanisms of processes that developed during the reconsolidation of contextual memory after a reminder depended on the 5-HT system and on the state of the memory at the time of the reminding procedure associated with the intensity of training.)
  3. The text of the article was again checked by experts who know English.

Sincerely,

Prof. Kh.L. Gainutdinov, Lieder scientist of Laboratory of Neuroreabilitation of Motor Disorders and prof. of Department of Physiology of Human and Animals of Institute of Fundamental Medicine and Biology of Kazan Federal University.

Reviewer 2 Report

- The subject addressed by the authors in the article: "Effects of thryptophan hydroxylase blockadeby p-chlorophenylalanine on contextual memory reconsolidation after training of different intensity" is interesting.
However, it probably requires further molecular research to be able to conclude "The obtain results allowed to conclude that the molecular processes ..."; this conclusion seems to be too speculative.

- The discussion is  insufficient, does not contain relevant arguments and explanations. A large part of the discussion (the beginning) does not bring significant information in relation to the work carried out by the authors, the discussion remains descriptive without really trying to explain the observed behaviours.

- The discussion needs a great deal of work done to it.

Author Response

Effects of tryptophan hydroxylase blockade by p-chlorophenylalanine on contextual memory reconsolidation after training of different intensity

Authors

Irina Deryabina, Viatcheslav Andrianov, Lyudmila Muranova, Tatiana Bogodvid, Khalil Gainutdinov*

Dear Reviewer,

Thank You for the thorough review. We consider Your comments.

- The discussion is insufficient, does not contain relevant arguments and explanations. A large part of the discussion (the beginning) does not bring significant information in relation to the work carried out by the authors, the discussion remains descriptive without really trying to explain the observed behaviours.

  1. We agree with you, the beginning of the discussion is not directly related to the work performed by us. It is unnecessarily large; there we wanted to show violations associated with serotonin deficiency. Following the recommendations of the reviewer, we have reduced this part.
    2. In the final part of the discussion, we tried to explain what processes change in response to memory reactivation during serotonin deficiency. (It could be assumed that serotonin in this case was necessary for the initiation of reconsolidation, or for its complete completion. The absence of the cumulative effect of blocking protein synthesis and serotonin synthesis might indicate different mechanisms (or ways) of consolidation and reconsolidation.)

- However, it probably requires further molecular research to be able to conclude "The obtain results allowed to conclude that the molecular processes ..."; this conclusion seems to be too speculative.

  1. We agree that such an assumption is this conclusion seems to be too speculative.
    We replaced it with a conclusion that is more consistent with the results.

(Mechanisms of processes that developed during the reconsolidation of contextual memory after a reminder depended on the 5-HT system and on the state of the memory at the time of the reminding procedure associated with the intensity of training.)

  1. The text of the article was again checked by experts who know English.

Sincerely,

Prof. Kh.L. Gainutdinov, Lieder scientist of Laboratory of Neuroreabilitation of Motor Disorders and prof. of Department of Physiology of Human and Animals of Institute of Fundamental Medicine and Biology of Kazan Federal University.

Round 2

Reviewer 2 Report

the authors have made efforts to improve the manuscript. the manuscript may, if accepted by the editor, be published in the IJMS journal.

Author Response

Effects of tryptophan hydroxylase blockade by p-chlorophenylalanine on contextual memory reconsolidation after training of different intensity

Authors

Irina Deryabina, Viatcheslav Andrianov, Lyudmila Muranova, Tatiana Bogodvid, Khalil Gainutdinov*

Dear Reviewer,

Thank You for the thorough review.

Sincerely,

Prof. Kh.L. Gainutdinov, Lieder scientist of Laboratory of Neuroreabilitation of Motor Disorders and prof. of Department of Physiology of Human and Animals of Institute of Fundamental Medicine and Biology of Kazan Federal University.

This manuscript is a resubmission of an earlier submission. The following is a list of the peer review reports and author responses from that submission.

Round 1

Reviewer 1 Report

The authors investigate the role of serotonin  in reactivated contextual memory. Behavioral tests are performed on snails and synthesis of serotonin is inhibited by the administration of PCPA. After testing different protocol conditions, the authors conclude that serotonin has a key role in the reconsolidation of contextual memory.

-English is not always fluent and could be ameliorated.

-The concept of "memory of different strenghts" is not clear. How can we measure the strenght of a memory? What about the tests with snails? This point should be better duscussed.

-Results are clearly presented and discussed but I have a major concern about the protocol:

Authors used PCPA to inhibit serotonin synthesis but I could find no reference concerning brain serotonin levels in snails. Even going back to previous works, the only data I found concerned other animal models (mainly mice and rats). I think authors should refer to snail protocols and experiences as different models might have completely different distribution of PCPA and consequently different effects and timing on brain serotonin levels. If not previously done, authors should measure serotnin brain levels after treatments.

- Finally, peripherally injected PCPA could have a role on snail behavior other than decreasing serotonin brain levels. As the authors discuss in the introduction, serotonin has a plathora of effects. These effects are not only restricted to the central nervous system. Could defensive reactions be impacted by the peripheral effects of serotonin? This should be discussed.

Reviewer 2 Report

A similar work has already been published in Front Pharmacol under the reference: Front Pharmacol. 2018 Jun 12; 9: 607. doi: 10.3389 / fphar.2018.00607.